# Responding to Global Challenges through Education: Entrepreneurial, Sustainable, and Pro-Environmental Education in Nordic Teacher Education Curricula

**Jaana Seikkula-Leino** [1,2,*], **Svanborg R. Jónsdóttir** [3], **Marcia Håkansson-Lindqvist** [2], **Mats Westerberg** [4] **and Sofia Eriksson-Bergström** [2]

1 Renewing Learning and Collaboration, Tampere University of Applied Sciences, 333520 Tampere, Finland
2 Department of Education, Faculty of Human Sciences, Mid Sweden University, SE-85170 Sundsvall, Sweden; marcia.hakanssonlindqvist@miun.se (M.H.-L.); sofia.eriksson-bergstrom@miun.se (S.E.-B.)
3 School of Education, University of Iceland, 105 Reykjavík, Iceland; svanjons@hi.is
4 Department of Social Sciences, Technology and Arts, Luleå University of Technology, SE-97187 Luleå, Sweden; mats.westerberg@ltu.se
* Correspondence: jaana.seikkula-leino@tuni.fi or jaana.seikkula-leino@miun.se

**Abstract:** The United Nations' Sustainable Development Goals (SDGs), and the European Union's strategies both set goals for solving environmental challenges faced by societies and communities. As part of solving these challenges, both the UN and the EU stress the development of entrepreneurial and innovative education. Teacher education plays a crucial role in these efforts, since teachers and teacher educators have a significant impact on educating citizens far into the future. In this research, we studied how Nordic (Finnish, Swedish, and Icelandic) primary teacher education curricula involve entrepreneurial, sustainable, and pro-environmental education. For this study, the authors analyzed the B.Ed. curricula of three academic teacher education institutions in Spring 2021. We used qualitative content analysis as our research method. According to the results, all three curricula incorporated both entrepreneurship education and sustainable development to some extent, although often not very explicitly. Given the urgency of problems such as global climate change, the educational goals and contents in these curricula related to entrepreneurial education and sustainable development are very limited. The idea of integrating environmental/sustainable and entrepreneurship education could be promoted in the future more explicitly, with these interdisciplinary educational themes emphasised more strongly in the curricula and education policies.

**Keywords:** entrepreneurship education; sustainability; sustainable education; teacher education; curriculum; Nordic education

## 1. Introduction

According to the United Nations, science, technology, and innovation (STI, which includes entrepreneurial thinking) have long been recognized as one of the main drivers behind productivity increases and a key long-term lever for economic growth and prosperity [1,2]. In the European context, sustainable development and entrepreneurship are put forward as important areas for education [3–5].

Transitioning to environmentally sustainable societies also has the potential to create millions of jobs, which requires dynamic entrepreneurial competencies. However, this will require bold action to invest in people's capabilities to increase their productivity and realize their full potential [6]. Because there is great potential in entrepreneurship and entrepreneurial activity, governments have already made significant investments in innovation, entrepreneurship, and entrepreneurial education programs [7]. This is also supported by policy initiatives and economic evidence, such as those published by the Organisation for Economic Co-operation and Development (OECD) [8] and World Bank [9].

However, our existing knowledge of how entrepreneurial activities could contribute to the SDGs remains limited [2]. Many of the existing studies are conceptual and focus mainly on an individual or on organisational-level factors [10,11]. Thus, we need more research on how pro-environmental and entrepreneurial behavior can be realized in wider learning contexts [12]. In addition to this, the study of the phenomenon should be more in-depth. The recent academic debate is both too descriptive, optimistic and too indefinite integrating the research areas of entrepreneurship and sustainability. For example, according to Filsher et al. [11], despite the increasing trend towards sustainability-related entrepreneurship literature, only six of their 21 reviewed papers published after 2015 address the SDGs. In most of these papers, the SDGs are mentioned as an introductory example and not examined in depth. Furthermore, it is worth remembering that these concepts can also take on opposite meanings in the general debate. For example, the activities of companies can also pose environmental risks, and therefore entrepreneurship and sustainable education might be seen as contrary. These connotations can contribute to how "new concepts" of education, e.g., sustainable and entrepreneurship education, are understood and implemented in education and how their potential is seen more broadly.

At present, international and national strategies promote leading changes in education curricula including sustainable development and entrepreneurship. However, high-level strategy is a different matter from what is happening on the ground. As an example, for many areas of higher education, for example business, economics, and the natural sciences, the areas of sustainable development and entrepreneurship are taken for granted as part of the curriculum. Although, in the case of teacher education this may not be the case. The EU Commission Report in 2011, Entrepreneurship Education: Enabling teachers as a critical success factor states that in many EU countries, there is a large gap between the implementation of entrepreneurship education in primary schools and teaching entrepreneurship education to preservice teachers in higher education institutions. Therefore, the report stresses that the core skills linked to entrepreneurship are seldom a priority in initial teacher education. However, the report was published ten years ago. Has anything happened since then?

We also want to focus in particular on Nordic education. Nordic societies and their educational strategies naturally emphasize the responsibility and freedom of learners to develop a better society. Therefore, it could be expected that the Nordic curricula and their teacher training would proactively take these educational objectives into account. Or do we just think so? Have the Nordic countries been able to take advantage of this privilege in the planning and implementation of modern education, which may not always be possible at the global level? Some studies show that there are challenges in integrating entrepreneurship and sustainable education in curricula and teacher education. On the other hand, there are differences between universities [13–15]. However, none of these studies simultaneously look at both entrepreneurship and sustainable development in teacher education. Therefore, this study explores how teacher education in three selected cases in three Nordic countries, Finland, Sweden and Iceland, are actually designed with regard to entrepreneurship education and sustainable development. This will help us understand the future, as some of the teachers currently being trained will teach up to the 2050s. Today's curricula, therefore, have far-reaching implications for the future.

To clarify, we refer simultaneously to both mainstream concepts of *entrepreneurial education and entrepreneurship education* whilst investigating the curricula in education: Entrepreneurship education is more than content. It is also considered as a method and practice for learning [14,16]. Furthermore, in our study, we understand *pro-environmental* behaviour as "behavior that consciously seeks to minimize the negative impact of one's action on the natural and built world" [17], (p. 240). To cultivate such behaviors requires participatory teaching and learning methods that motivate and empower learners to change their behavior and become willing and capable to take action for *sustainability* which means meeting our own needs without compromising the ability of future generations to meet their own needs [18].

As in the definition of entrepreneurship education, in this study we refer simultaneously to the concepts sustainable, pro-environmental, and environmental, as if they were equivalent, even if they are not. This is due to the fact that in general education, these concepts are used freely and as corresponding to each other, even in curricula design. Therefore, if we want to study the phenomenon itself, it makes sense for us to look at how educators in general have incorporated any of those concepts related to others in their curricula writing.

The paper is structured as follows: Firstly, in the literature review section, we summarise the shift towards entrepreneurial and sustainable societies and stress teacher education and curricula in the context. Secondly, we provide an overview of the study and the methods based on content analysis. Thirdly, we present the results from teacher education curricula analysis. Finally, we discuss our key findings from the Nordic countries, and suggest some directions for the future.

## 2. Literature Review

### 2.1. Entrepreneurship Education as an Engine for Promoting Sustainable Transformation

The purpose of entrepreneurship education is to educate students to take more responsibility for themselves and their learning, to try to achieve their goals, to be creative, to discover existing opportunities, and to cope in a complicated society [19]. Another aim is for them to take an active role in the labor market and consider entrepreneurship as a natural career choice. Entrepreneurship education involves developing behavior, skills, and attributes, applied both individually and collectively, to help individuals and organizations of all kinds to create, cope with, and enjoy change and innovation [16,20–23].

Research on entrepreneurship education is based, in large measure, on a conceptual understanding of entrepreneurship and learning [4,24,25]. Some researchers have focused on identifying and analysing the core pedagogy of entrepreneurship education, characterizing it as emancipatory pedagogy, where the aim is to empower learners to become independent, creative, and active participants in society [10,19]. Prior research suggests that developing entrepreneurial mindsets is a key engine of growth and a must for sustainable development—e.g., in promoting the UN SDGs—and social cohesion, both locally and regionally, e.g., [26]. There is growing evidence of the impact of entrepreneurship education: studies indicate that entrepreneurship or entrepreneurial education can increase youths' entrepreneurial intentions and knowledge; stimulate their creativity, collaborative abilities, and self-confidence; and enhance their learning of other subjects, e.g., [27]. By developing an entrepreneurial mindset in societies, we can open an arena in which pro-environmental and sustainable solutions could be created more innovatively and co-creatively. The added value of entrepreneurship education has been understood in children's and youth education.

In addition to this, sustainable development plays a significant role in today's entrepreneurship academia and practical discussions. Shepherd and Patzelt [28] define sustainable entrepreneurship as follows: Sustainable entrepreneurship is concerned with the preservation of nature, life support, and community in the pursuit of perceived opportunities to create future products, processes, and services for profit, where profit is broadly defined to include both monetary and nonmonetary benefits to individuals, the economy, and society.

However, the link between entrepreneurship education and sustainable development has not always been recognized. According to Hermann and Bossle [29] entrepreneurial abilities such as foresight, problem-solving skills, and interdisciplinarity have been neglected in sustainability education. However, although entrepreneurship and sustainability education have separate learning objectives that are unlikely to be combined, major thematic similarities in interdisciplinary entrepreneurship and sustainability education can be identified. They are, e.g., innovation design, entrepreneurship ecosystem support, and corporate/organizational aspects. Furthermore, as Hsu and Pivec [30] argue, integrating sustainability into entrepreneurship education, including comprehensive plans in curricula,

could have more potential than we are aware of in education development and promoting major goals, e.g., SDGs. As a matter of fact, Edokpolor [31] stresses the relationship between entrepreneurship education and the core values of sustainable development.

It seems that the potential and challenges of integrating entrepreneurship education and sustainable development have been recognized, and entrepreneurship education and education in sustainable development appear to be just out of reach of the school curricula in most countries, which have yet to consistently teach children to find resources to put their ideas into action. The focus on action competence as an aim of sustainability education strongly resembles the focus of entrepreneurship education on fostering entrepreneurial competence through creativity and action [32]. However, the development of societies and education takes decades. Achieving the EU's Green Transformation goal of climate neutrality by 2050 requires that the relevant competences and skills be developed by 2030. This also presupposes the integration of different competence frameworks, (e.g., EntreComp and GreenComp) and the development of corresponding educational concepts, e.g., curricula, at various levels of education. Within this context, entrepreneurship education has a central role as an engine for promoting sustainable transformation. This is also highlighted in Education 2030 by OECD [33] as environmental, economic, and social transformation and a proposed associated learning framework that encompasses disciplinary ideas, cross-cutting concepts, and social and economic practices. Thus, there is an increasing emphasis on entrepreneurship education in the field of education. The potential of entrepreneurship education has also been understood in teacher education, as teacher educators train future teachers who have a long-term impact on the future [24].

### 2.2. Sustainable and Pro-Environmental Education in the Nordic Context

*Education for sustainable development (ESD)* is meant to inculcate competencies in critical and creative thinking, imagination, and collaboration. Students need these skills to tackle the complex social, environmental, and economic issues and challenges of the modern world [18]. Instead of learning traditional discipline-focused areas, we also need to create opportunities for multidisciplinary and even phenomenon-based learning, e.g., [34], in which learners apply different perspectives to study real-world problems. To understand and solve problems related to climate change, for example, knowledge is needed from different subjects such as natural sciences, geography, psychology, economics, mathematics, and history.

Jóhannesson et al. [35] identified core characteristics that indicate sustainable development in curricula. These researchers encouraged a holistic view of sustainable development, looking at economic, environmental, and social factors as integrative entities. The characteristics were meant to reflect the goals of the United Nations Decade of Education for Sustainable Development 2005–2014 and research on environmental education and education for sustainable development.

Cars and West [36] argue that ESD can be understood as an educational ideology that came about by adding a developmental component to environmental education. Here, there are three overlapping and sometimes conflicting spheres of sustainable development— natural ecology, social issues, and economic factors. Further, ESD is meant to help people to develop attitudes, skills, and knowledge that supports them in making informed decisions that benefit themselves and others and to act upon these decisions [37]. ESD is cross-disciplinary by nature [38] and could be a catalyst for social changes and social transformations to greater equity. Thus, ESD can be seen as an application of critical pedagogy [36]. The core of sustainability education is to empower learners with the competence for action [39].

A critical distinction has been made between education for sustainable development (ESD) and *sustainability education (SE)*. ESD is defined as education that includes first and possibly second order changes, where SE is more radical and includes third order changes [40,41]. First order changes and learning take place within accepted frameworks, leaving basic values unexamined and unchanged. Second order changes involve critically

reflective learning, where assumptions about the world are challenged. Third order changes are deconstructive and reconstructive, involving a deep awareness of versatile worldviews and ways of doing things, encouraging radicality and action [40,41]. ESD may be seen as a necessary journey towards SE, and *entrepreneurship education (EE)* could be a part of constructive steps towards SE. EE includes affordances that contribute to inculcating analytical and creative competences. These comptetences are needed for responsible action and provide cognitive elements (knowledge and understanding), emotions (identifying needs and problems), and action [19,40].

A comprehensive Nordic report on ESD maps education for sustainability in the Nordic countries, scrutinizing laws, regulations, national curricula, curricula of teacher education institutions, research, and reports in pre-, primary, and secondary education [40]. The report shows that the word *sustainability* is not mentioned once in the actual law on educational acts in the Nordic countries. However, the laws address issues such as democracy, human rights, equality, and respect for nature, which are all elements of sustainability education. The authors of the report also indicate that sustainability is often mentioned explicitly at the level of national curricula, special reports, and strategy papers from the Ministries of Education. One example is the Icelandic national curriculum established in 2011, which identifies sustainable development as one of six pillars of education. Sustainability is one of the core values of the Finnish national core curriculum for basic education, where it is mentioned almost 200 times. According to the Jónsson report [40], ESD has been present in the Swedish national curricula since 1994 and was written into law in the Higher Education Act of 2006. The report, however, presents a somewhat confusing picture. For example, in Iceland, educational policy seems to vacillate between strongly emphasizing sustainability, and not emphasising it at all. Iceland's law on compulsory education from 2008 has very little to say directly about sustainability. Sustainability in educational policies in Finland, Norway, and Sweden builds on a long tradition of environmental education and has been more consistent than in Iceland or Denmark. However, the Finnish, Norwegian, and Swedish educational acts fail to mention sustainability explicitly. Though 'sustainability' or variants appear almost 200 times in the Finnish national core curriculum, "the incorporation of sustainability as an educational aim or subject is often superficial" (p. 64). The authors of the report conclude that human existence has become less sustainable, and that conventional education is part of the problem and needs to be drastically redesigned [40].

### 2.3. What Could Be the Opportunities of Nordic Teacher Education?

Although the importance of teacher education has been emphasized, EE still seems to be a moderately overlooked theme in teacher education across our three countries [24,42]. Similarly, while there have clearly been attempts to include environmental education and research ideas in teacher education, these are not yet bearing much fruit – and indeed, a number of studies from around the world suggest that environmental education is not easy to fit into general teacher education programmes [13,43–48]. In summary, research emphasizes that strengthening both EE and sustainable education in teacher education would have more added value than we might think.

In the Nordic context, the development of future competencies for pro-environmental behavior may have unique potential, as the Nordic countries have a long tradition of advancing the goals of sustainable development at the national level and have been assessed as among the most SDG-ready countries. A renewed Nordic cooperation programme targeting the 2030 Agenda has the potential to help the Nordic countries become even more successful and effective, and to bring added value to the work done internationally, e.g., [49]. Furthermore, the Nordic model of education is based on national education systems that build on specific local values and practices but are influenced by international goals. Equity, participation, and welfare form the ideal Nordic model, which places value on shifting education towards more innovative, co-creative, and pro-environmental activities. The Nordic education model could be used more widely in global education de-

velopment [50], as it provides an ideal "platform" to develop and test entrepreneurial and pro-environmental education initiatives. At the same time, teacher education also provides another platform for designing and implementing modern teaching, as traditionally, new trends in education first come to the fore. Teacher education institutions are even required to act as leaders in educational development in many countries. The Nordic education model and its teacher education could have more added value and impact than we think. Therefore, our research profiles how both sustainable development and EE have been taken into account in Nordic teacher education.

### 2.4. Curriculum as a Tool for Educational Change?

The definition of a curriculum assumes that: (1) a curriculum lists the courses or programs that should be offered; and (2) a curriculum is highly experiential—it demonstrates, both indirectly and directly, the abilities and skills that the individual should achieve [51]. Curricula reflect societal values and valuations. Thus, curriculum reform springs especially from a social desire for change—e.g., to realise entrepreneurial and sustainable education— and in this case, it is directed by values and ideological and political aims. Ideas concerning the 'right knowledge' and the distribution of power steer the reforms and activities [15]. General social trends and challenges such as globalisation, climate change, technological development, and the needs of labor markets direct the objectives of education and therefore also steer curricular reforms [15,52].

Traditionally, curriculum has been seen as belonging to the primary and secondary education context. Discussions and research related to higher education curriculum have sometimes been considered as questioning the autonomy of higher education institutions [53,54]. There has been little research on higher education curricula, and what there is has limited itself to specific fields [42]. Naturally, it is problematic if curricular concepts and theories coming from a primary/secondary school context are applied straightforwardly to higher education [54]. However, as Barnett and Coate [55] argue, the curriculum should be one of the core concepts used when developing higher education from research and pedagogical points of view.

It is important to examine curricula because they form the most important administrative documents that determine the content of training [42,56,57]. If entrepreneurship and sustainable education are to be systematically developed in the education system, it must be done via curricula. By looking at the three curricula documents of higher education institutions in Finland, Sweden, and Iceland, we can determine the direction of entrepreneurship and sustainable education in Nordic society. Moreover, this will indicate how these up-to-date themes of education are proceeding at some level globally. Besides our results, we will include ideas on how entrepreneurship and sustainable education could be developed in the future within teacher training, and how curriculum design can be developed more deliberately. Our study focuses especially on teacher education curricula, and more narrowly on primary teacher education on bachelor level because in our research countries, all teacher trainees receive at least bachelor level training.

Higher education institutions are independent developers of education. This element factors how they have wanted to or been able to include entrepreneurship and sustainable education in their teacher training curricula. The previously described background of the educational needs has also provoked our targets in our study, especially in teacher education, where elements such as entrepreneurship and sustainable education are integrated into curricula more or less in line with international and national strategies and documents. Therefore, we want to study how Nordic teacher education curricula have adapted entrepreneurship and sustainable education. We also want to broaden our understanding of whether "new winds" of education have been considered in this Nordic region, as might be expected.

As a summary from our literature review: Since we study how EE and sustainable development are reflected in the Nordic teacher education curricula, we emphasize curriculum research as the central administrative documents that guide the development

and implementation of instruction. We also emphasize teacher education because it has far-reaching implications for the future. Furthermore, we also focus on research in Nordic education because we believe that Nordic education can be a pioneer in the field. Therefore, our findings can give preliminary indicators into where education is moving onto. Based on this, the main themes of our research are entrepreneurship education (EE), sustainable development, environmental education, teacher education, curriculum, and Nordic education. Furthermore, other related concepts, such as entrepreneurial learning, and SDGs have been presented in the literature review to broaden the understanding of our study and its methodological choices. Next, we will describe our research question and the chosen methodology of our study.

## 3. Research Question and Methodology

To investigate how and to what extent the Nordic countries incorporate entrepreneurial and sustainable education into their teacher education curricula, we pose the following research question:

- How does entrepreneurial, sustainable, and pro-environmental education emerge in Nordic (Finnish, Swedish, and Icelandic) primary teacher education curricula?

For this study, the authors analysed the B.Ed. curricula of the three academic teacher education providers in Finland, Sweden and Iceland, in April–May 2021. In the analysis, we looked for specific types of curricular topics or subjects referring to sustainability and entrepreneurship education (EE), (and concepts related), since we consider these issues are essential in two respects: First, we need to have some evidence to guide understanding of where these topics are situated in the framework documents (e.g., indicate the extent to which the learning framework supports "environmental, economic and social transformation" (28); and, second, to draw attention more fully—especially in conclusion—to strategies and future curricula design to address the observed gaps.

The curricula were obtained online. Qualitative content analysis was the research method used to interpret the content of text data through the systematic classification process of coding and identifying themes or patterns. Content analysis is usually used with a study design whose aim is to describe a phenomenon [58]. The curricula were read first generally, and then reflectively, with the aim of finding explicit and inexplicit references to the following concepts: entrepreneurship education; entrepreneurial learning/skills/competencies; innovative/creative learning/education; sustainable education, environmental education; SDG. The concepts were then identified and analysed separately. A somewhat similar study was found in this area. For example, Seikkula-Leino et al. [15] studied entrepreneurship education in teacher education curricula. As in this study, we also utilized a similar content analysis technique found to work in that study. At this point, we felt it was essential to analyze the curricula first because they are the primary documents that guide the implementation of teaching. Therefore, we did not yet proceed with qualitative interviews, for example, or quantitative surveys.

In analyzing the material, we thoroughly reviewed the descriptions of the degree programs and their courses. We analyzed the objectives, contents, expected results, and course evaluation criteria. In addition, we reviewed the course learning material. A broader conceptual bank, as previously described, was to support the evaluation, e.g., the concept of EE is often not explicitly used but is referred to by other concepts (such as creativity and innovation). The concept bank made it possible to evaluate the material comprehensively. Suppose the main concepts used in this study are directly recorded in the title and objectives of the course. In that case, we consider it more important than the fact that, for example, subject teaching uses primary education school material, such as the national curriculum, which includes entrepreneurship and sustainable development. So we focused primarily on what the goals and contents are in the bachelors' programs of our study. However, the references were not always so easy to find. Therefore, as described, we also extended looking at the learning material used to get inside the phenomenon somehow.

- At a minimum, we assessed each curriculum for the following characteristics: expediency, authenticity, relevance, administrative approval within the organisation, accuracy, and objectivity [59]. When it was clear that a curriculum steered the organisation's actions, all these characteristics were present. However, when curricula were obtained online, authenticity was an important factor to consider.
- Overall, this type of even minor pilot study with three universities is a good place to start looking at how these global aims are manifested in central educational documents. As Cohen et al. argue [59], the generalizability of single experiments (e.g., case and pilot studies) can be further extended through wider replication or multiple experiment strategies, allowing single pilot studies to contribute to the development of a growing pool of data, and allowing the key findings to be more broadly generalized in the future.

Below we describe in detail of the case examples of teacher education in Finland, Sweden, and Iceland. This is followed by analysis of their curricula, discussion, and conclusions.

## 4. Case Study Overview

Next, we briefly present the case of educational organizations, and their country context in teacher education, involved in the study from which the three curricula cases were collected and analyzed to understand how entrepreneurship education and sustainable/pro-environmental education are involved in teacher education. We chose only one teacher training organization from each country because mainly one university provides the most teacher training in Iceland.

### 4.1. Finnish Teacher Education: University of Turku, Faculty of Education, Teacher Education, Rauma Campus

In Finland, teacher training is arranged by universities and vocational institutes of higher education. They train pre-school, classroom, subject, special education, and vocational teachers. Academic teacher education is offered by 12 higher education units and their 13 teacher training schools [42]. Higher education institutions decide independently on the contents of teacher education, and emphasise the link between teaching and research. All teacher education also involves pedagogical studies and guided teaching practice. These are realized in the universities' own schools for teaching practice or other schools nominated for the purpose [60].

The teacher education curriculum at the University of Turku, Rauma Campus program includes studies in educational science, teaching internships, and multidisciplinary studies in the subjects and subject areas taught in basic education. The graduating classroom (primary school) teacher is prepared for both independent work and interprofessional cooperation as a teacher and educator. The aim of the degree program is the ability to meet and teach students from different cultural backgrounds and abilities. Most of those who graduate as class teachers work in teaching positions. However, the training also equips for administrative, planning, research and development tasks in the field of education. In Rauma Campus is also a teacher training school educating pupils, students, and student teachers [61].

### 4.2. Swedish Teacher Education: Mid Sweden University, Department of Education, Campus Sundsvall

The current teacher education system was introduced in 2011 as an outcome of the many official reports by the Swedish government that examined Swedish teacher education. Swedish teacher education has been fundamentally reformed several times since the Second World War. The teacher education that exists in Sweden today emphasizes subject knowledge, and thus gives the academic education ideology more space than before. All teacher education is run by a college or university and is nationally established and governed in accordance with the Higher Education Act (SFS 1992:1434) and the Higher

Education Ordinance (SFS 1993:100), as is all higher education. Twenty-eight different universities offer teacher education programs.

In Sweden, teacher education is organized through different programs that correspond to different ages of pupils in school. The Swedish education system includes preschool to high school, for children from 1 to 16 years old. The current teacher education system has reintroduced a clear division with different programmes of education for different teacher categories: grades 1–3, grades 4–6, grades 7–9, and upper secondary school. The argument is that pupils of different ages require different kinds of knowledge and skills. The programs include different subject areas such as educational science, didactics, studies in specific subjects, and internship education. The internship education requires the student, under supervision, to plan and carry out activities in the school. These 30 European Credit Transfer System (ECTS) credits, in one term, are distributed over various shorter periods throughout the program. Teacher education is vocationally oriented and aims for the student to develop a scientific and pedagogical approach, theoretical understanding, practical knowledge, as well as to develop as a person.

At Mid Sweden University, the teacher education is organized with a campus/distance learning model in which students conduct their studies in their hometown and spend only four to five weeks per semester on campus with three to five days of scheduled activities per week.

*4.3. Icelandic Teacher Education: University of Iceland, School of Education*

In 2008, teacher education in Iceland changed from a three-year B.Ed. degree to a five-year Master's degree (Act nr. 87/2008). To become licensed teachers before 2008, most students enrolled in three-year programmes at the Iceland University of Education (later the School of Education (SoI) at the University of Iceland), or the University of Akureyri. Those who already had a B.Ed. degree kept their license, but many have chosen to add a master's degree. Vocational education teachers require 60 ECTS in teacher certification studies in addition to a final diploma in their vocation (e.g., master craftsperson). The premise behind adding the master's level in 2008 was that teachers needed to be involved in research and the development of knowledge and thus strengthen their professional self-image [62]. According to the current law governing teacher education in Iceland (Act nr. 95/2019), student teachers must complete a master's degree and have both general competence as well as a specific competence such as completing at least 90 ECTS in a specific school subject.

The fundamental B.Ed. degree at the SoI is a 180 ECTS programme of academic and practical studies for those who intend to teach grades 1–5 in compulsory schools. The goal is for students to have knowledge of children's development, how they learn and communicate, literacy and teaching reading, and use of language. Emphasis is placed on the main learning areas and subjects at the primary level [63]. Theoretical content and field practice are woven into courses; this includes the interaction of theory and practice. Since the law requiring master's level education took effect in 2011, there have been contradicting pulls and conflicts in the development of teacher education in Iceland [64]. Conflicts have emerged between teacher education programs and the State about who is responsible for teacher education and what it should contain. Within the SoI itself, the development of the programme has involved arguments and conflicts between a focus on specialisation versus a focus on breadth of knowledge [64].

After the three case presentations described above, we look at the study results: an analysis of curricula from these three teacher training units.

## 5. Results

The following Table 1 describes the outcomes of curricula analysis of three universities step by step. Our data show that EE and sustainable development are taken into account to some extent in teacher education curricula in the Nordic countries. All in all, the teaching units related to entrepreneurship and sustainable education are part of, for example,

subject studies or optional studies. The goals and contents of biology and other science, for example, include the starting points for environmental education, thus including sustainable development. Moreover, our results indicate that teaching these themes is not stressed at any particular year level.

Considering how much these themes are discussed today, these educational goals and contents are scarce. In Finland, entrepreneurship/entrepreneurial and sustainable education are widely approached, primarily through the national basic education core curriculum. These studies are available in both compulsory and optional studies. Education is also provided in the preparation and implementation of the curriculum. Thus, the curriculum itself seems to assume that future teachers will be somehow trained to implement both entrepreneurial learning and sustainable development. On the other hand, the aims and contents of the teacher education curriculum do not explicitly mention this elsewhere.

In Iceland, entrepreneurship and entrepreneurial education are not visible as distinctive elements in the University of Iceland B.Ed. program. This element "entrepreneurial learning" is mentioned once in the compulsory school core curriculum, however initiative is often mentioned there, often in relation to creativity and or independence. Neither were found in the B.Ed program for primary education teachers. The sustainability concept (34 times) is also clearly visible in the Icelandic compulsory school general curriculum and so is creativity (38 times in different compositions). Similarly to the Finnish case, the conclusion is that the aims and contents of the B.Ed. teacher education curriculum for primary school level at the School of Education (SoI) only mention these concepts in optional studies.

In Sweden, entrepreneurship/entrepreneurial education is not an explicit element in primary education for teacher students. Although entrepreneurship is mentioned twice in the general curriculum, one instance refers to entrepreneurship as a fundamental goal and task of the schools. The other instance refers to the 7–9 school level, where this concept is not included in the program plan for student teachers. Neither is the concept of sustainable development found explicitly in the primary teacher education plan. However, the concept of sustainable development is explicit (38 times) in the Swedish curriculum for the national core curriculum for primary education that teacher education utilizes in their education. Here, this concept is general for the fundamental goals and tasks of the schools and for the primary level of school. The concept of creativity/creative ability is seen in the curriculum, but it is not included in the goals for the primary education program plan. Thus, the Swedish case appears to be in line with the Icelandic and Finnish cases.

**Table 1.** The Content Analysis of Nordic Teacher Education Curricula with Three Case Examples.

| Curriculum Case Studies: Country Examples from Primary Teacher Education, Bachelor's Degree Programme/Steps and Outcomes by Content Analysis | (1) Curriculum Case Finland: *Class Teacher Education (BA), University of Turku, Faculty of Education, Class Teacher Education, Rauma* | (2) Curriculum Case Sweden: Mid Sweden University | (3) Curriculum Case Iceland University of Iceland, School of Education, Primary Education B.Ed. |
|---|---|---|---|
| The content was read several times in order to build an overall picture of the curriculum | General notes:<br><br>The programme (180 ECTS) consists of studies in educational science, practical training and studies that provide pedagogical skills required for positions in primary education.<br><br>The aim of the degree programme is to educate well-qualified academic professionals, active future makers, critical and ethically responsible experts and researchers in the educational field. At the core of a teacher's work lies the understanding and supporting of a child's and group's development. The dialogue between theory and practice takes place particularly during teaching practice periods, which offer a holistic view of a teacher's work.<br><br>*The objective of the programme is to develop the following areas of competence:*<br><br>1. Communicative competence. Student is able to act collaboratively and is capable of communicating in different interactional situations.<br>2. Pedagogical competence. Student knows the basics of curriculum planning and is able to plan, implement, evaluate and develop learning processes and learning environments.<br>3. Intellectual competence. Student has basic knowledge of education as a science and contents of the multidisciplinary subjects taught in basic education. Student base his/her actions and professional development on scientific thinking. Student understands the principles of the societal and cultural basics of childhood.<br>4. Ethical competence. Student is able to identify and analyse his/her actions from an ethical viewpoint and act in accordance with ethical principles. | General notes:<br><br>The programme (240 ECTS) consists of studies in educational science, practical training and studies that provide pedagogical skills required for positions in primary education, year 1–3 and year 4–6:<br><br>*Primary School Teacher Education Programme in Pre-school Class and School Years 1–3 (Lärarutbildning-Grundlärare med inriktning mot arbete I förskoleklass och grundskolans årskurs 1–3, 240 hp)*<br><br>*Primary School Teacher Education Programme, Years 4–6 (Lärarutbildning-Grundlärare med inriktning mot arbete igrundskolans årskurs 4–6, 240 hp)*<br><br>*Aim*<br>*The aim of the program is to for the student through theoretical, scientific and practice-based studies support students with the knowledge and skills needed to be able to work independently as teachers in year 0. 1–3.*<br>Subject and subject-didactic studies totalling 165 ECTS in Swedish, Mathematics, English, Civics, Natural science and Technology. For Swedish and Mathematics, 30 ECTS are required and for English 15 ECTS. Further, *General Education*, 60 ECTS and 60 ECTS Practice-based training. Of the subject and subject-didactic studies 15 ECTS must be subject-related practice-based training | General notes:<br><br>The programme is a three year 180 ECTS BEd studies (of five obligatory with two years on master's level to get a teaching permit). It consists of academic and practical studies emphasising teaching and learning of grades 1–5.<br><br>The programme focuses on:<br><br>• Childrens' development, communicative competence and use of language, learning to read, how to teach reading as well as first language learning.<br>• Rich emphasis is on the importance of collaboration with parents and the main learning subjects in compulsory school.<br>• The studies are conducted in close collaboration with the field (schools etc.) and are integrated with theoretical preparation for further studies and work in compulsory school.<br><br>The studies consist of the following elements and subjects:<br><br>• Icelandic<br>• Literacy and teaching reading<br>• Developmental and learning psychology<br>• Methodology and educational research<br>• Mathematics<br>• Teaching primary school students<br>• Curriculum and assessment<br>• Childrens' literature<br>• Speech and written text<br><br>Responses/solutions to challenges in play and learning<br>By reading the obligatory courses and bound electives, the following objectives seem to guide the studies. The competences thus extracted are (such analysed – not presented directly):<br><br>1. Knowledge of curricula guiding primary school education in Iceland.<br>2. Pedagogical competence. The teacher student knows the basics of curriculum planning and can plan, implement, evaluate and develop versatile learning processes and learning environments for different learners.<br>3. Intellectual competence. Student has basic knowledge of developmental psychology and learning challenges. Has knowledge of basic contents and teaching strategies of fundamental subjects (Icelandic, reading, writing, mathematics) as well as of integration of subjects.<br>4. The student can identify his/ her professional working theory and can understand his/her actions in relation to personal and academic theories. |

**Table 1.** *Cont.*

| Curriculum Case Studies: Country Examples from Primary Teacher Education, Bachelor's Degree Programme/Steps and Outcomes by Content Analysis | (1) Curriculum Case Finland: *Class Teacher Education (BA), University of Turku, Faculty of Education, Class Teacher Education, Rauma* | (2) Curriculum Case Sweden: Mid Sweden University | (3) Curriculum Case Iceland University of Iceland, School of Education, Primary Education B.Ed. |
|---|---|---|---|
| The curriculum was read reflectively, the aim being to find explicit refers to concepts: entrepreneurship education; entrepreneurial: learning/skill/competencies; innovative/creative learning/education; sustainable education, environmental education, SDG | Explicit refers:<br>1. Elective study: Entrepreneurship education and entrepreneurial pedagogy in early learning, 5 ECT<br>2. Elective study: Advanced course in nature and environmental education, 3 ECT, (refers also to sustainable education and SDGs | Explicit refers:<br>Primary School Teacher Education Programme, and Years 1–3; Years 4–6 (Utbildningsplan för: Lärarutbildning–Grundlärare med inriktning mot arbete I grundskolans årskurs 4–6, 240 hp)<br><br>*Reflective reading–no references to environment; entrepreneurship; innovative learning. Same for years 1–3 and years 4–6.*<br><br>Values–the history of school, organization and conditions, values, including basic democratic values and human rights. (p. 2) | Explicit refers:<br>No explicit references are made to entrepreneurship (or other forms of entre- enter- entrepreneurial) or action competence. Hardly any direct references to creativity except in two titles of obligatory courses and ways of working but not as a competence. One bound elective course (visual arts) has creativity as a competence aim. Innovative or innovation is not to be found.<br><br>Sustainability is only mentioned once as an element within one course (5 ECTS Integrative and creative work) in the third year. In the same course critical and creative thinking is presented as a core thread. |
| The curriculum was read more reflectively and analytically, the aim being to find inexplicit refers to concepts/themes: entrepreneurship education; entrepreneurial learning/skills competencies; innovative/creative learning/education; sustainable education, environmental education, SDG | (1) Orientation to Teaching Practice in Elementary School, 4 ECTS: National Core Curriculum for Basic Education as a learning material which includes several refers to Multidisciplinary Learning Modules (e.g., Working life and entrepreneurial competence; Participation, influence, and building a sustainable future) for integrating learning and for increasing the dialogue between different subjects. Furthermore, the subject studies in core curricula involve multidisciplinary learning themes.<br>(2) Didactic Teaching Practice in Elementary School, 8 ECTS: National Core Curriculum for Basic Education (see the description above)<br>(3) Multidisciplinary Stud. in the Subjects and Cross-Curricular Themes Taught in Basic Educ.<br><br>National Core Curriculum for Basic Education as a learning material (see the description above).<br>Other learning material to support local/school based curriculum design and its implementation which have aims and contents for entrepreneurship education and sustainability.<br>(4) Biology and Health Education, 4 ECTS: Some of the learning material involve topics such as environmental and nature protection.<br>(5) Introduction to Craft, Design and Technology in Primary Education, 6 ECTS: National Core Curriculum for Basic Education as a learning material (see the description above).<br><br>Other learning material of entrepreneurship education & arts and crafts education.<br>(6) Maker culture in arts and crafts, 3–4 ECTS, (inexplicit refers to entrepreneurial way of working) | In the national core curriculum:<br><br>*Curriculum for the compulsory school, preschool class and school-age educare, revised in 2018.*<br><br>(1) There are two references to entrepreneurship, entrepreneurial learning<br><br>Entrepreneurship is referred to in the general goals, and as a specific goal in the subject Civics.<br><br>(2) Sustainable development (38 references) to all school forms<br><br>Here the concept sustainable development is seen in the general goals as well as specific course goals.<br><br>Home and consumer education, Biology, Chemistry, History, Crafts and Technology.<br><br>Examples:<br>Chemistry–sustainable development is presented as a general goal, is stated implicitly in year 1–3 and explicitly in year 4–6 regarding grade levels.<br><br>Biology–implicit goal for 1–3, explicit for 4–6<br><br>Creativity<br><br>The concept creativity appears 15 times. Here, the concept is seen in the fundamental values an tasks, as well as subject goals for Art, Music and Crafts.<br>The concept creative ability occurs once. This is stated as an ability which pupils should acquire. | In the curriculum for Primary Teacher Education, Bachelor's Degree the signs of these issues or emphasis were either vague or only in some courses or not at all to be seen. Direct references to learning about or using the national core curriculum were to be seen in courses about: Reading and writing, information technology, mathematics, Icelandic, general introductory course, one course about visual arts, drama and music and in a course about curricular studies. However, none of these references were linked to or about the concepts and ideas we looked for.<br><br>Indicators of emphasis in the spirit of entrepreneurship education and sustainability education can be seen in the description of different courses. For example, in the only course that mentions sustainability there is an indication of ways of working that consider student experience and context: "An emphasis will be put on students' different experience and premises built on individualised learning and inclusion in a multicultural society where critical and creative thinking will be a core thread throughout the course". Similarly in the same course critical and analytical thinking and independent work is encouraged in the aim: The student will be able to discuss critically methods, ideas and issues in assessment of learning. In the bound choice obligatory course about *philosophy and ideas in education*, main ideas and ideologies in education are presented and might therefore cover ideas such as innovation and or entrepreneurship education and sustainability education although no direct mention is in the course's description.<br>These inexplicit indicators are all dependent on interpretation and guesswork and not clearly visible aims that can be associated with education for sustainable development or entrepreneurship education. It must be mentioned that teacher students can over the three years choose all in all ECTS from 59 courses (5 or 10 ECTS). In these courses some of them have explicit mentions of sustainability, initiative and creativity but they are not courses that all or most students choose and is up to chance which ones students choose. |

**Table 1.** *Cont.*

| Curriculum Case Studies: Country Examples from Primary Teacher Education, Bachelor's Degree Programme/Steps and Outcomes by Content Analysis | (1) Curriculum Case Finland: *Class Teacher Education (BA), University of Turku, Faculty of Education, Class Teacher Education, Rauma* | (2) Curriculum Case Sweden: Mid Sweden University | (3) Curriculum Case Iceland University of Iceland, School of Education, Primary Education B.Ed. |
|---|---|---|---|
| **Case Conclusions** | Entrepreneurship/entrepreneurial and sustainable education are widely approached, primarily through the basic education core curriculum. These studies are available in both compulsory and optional studies. Education is also provided in the preparation and implementation of the curriculum. Thus, it could be assumed that future teachers will be somehow trained to implement both entrepreneurial learning and sustainable development. On the other hand, the aims and contents of the teacher education curriculum hardly explicitly mention this elsewhere than in optional studies. | Entrepreneurship/entrepreneurial learning and sustainable development are approached in the general core curricula. However, in regard to the many goals and values, these concepts are few. The same is true for concepts referring to creativity. In the education program plans for primary education student teachers the concepts of entrepreneurship, sustainable development and creativity are not mention explicitly. This may mean that primary teacher student meet these concepts and content in specific courses. However, as prioritized elements of education, it is most likely that these concepts should be made more explicit to support teacher students in their work. | Entrepreneurship/entrepreneurial education is not visible as a distinctive element in the University of Iceland, School of Education, primary education B.Ed. program. This element "entrepreneurial learning" is once mentioned in the compulsory school core curriculum–however initiative is often mentioned there, often in relation to creativity and or independence. Neither were found in the B.Ed program. The sustainability concept (34 times) is also clearly visible in the compulsory school general curriculum and so is creativity (38 times in different compositions). |

Note. Adapted from University of Turku, Studying at the Faculty of Education [65].

## 6. Discussion

To investigate how and to what extent the Nordic countries incorporate entrepreneurial and sustainable education into their teacher education curricula, we aimed to study how entrepreneurial, sustainable, and pro-environmental education emerge in Nordic (Finnish, Swedish, and Icelandic) primary teacher education curricula. As this is a study of primary teacher education, which also aims to teach primary level education goals and contents, our results highlight this interdisciplinarity. For example, the contents of a teacher education curriculum often include references to primary-level materials, such as the national core curricula for primary education.

In our study results indicate that teacher education curricula have somehow inexplicitly refers to EE, entrepreneurial learning, sustainable education, sustainability, environmental, and pro-environmental education. However, none of the curricula clearly and explicitly address these themes in three Nordic countries in our study. Our data also demonstrate that these primary level teacher education objectives focus on foundational learning and pedagogical activities. Might it not be surprising to see the slight emphasis on higher-order phenomena like sustainable development or EE? However, it is also notable that these academic studies in teacher education always involve, e.g., traditional subject studies that have ancestral roots in disciplines created during hundreds of years throughout academia. Therefore, we could also question if it is enough today that, e.g., studies of math, literature, history, and sciences involve randomly and inexplicitly sustainable and entrepreneurial education? Therefore, we would challenge curricula design in education: Do these crucial and transversal areas of education need to be more explicit in modern teacher education curricula?

Strategies guiding education and teacher education have led to the integration of cross-cutting themes such as sustainable development and EE into teaching, e.g., [2,8,9]. However, there is a difference between what the steering documents say and what happens in practice. Higher education institutions are also autonomous in deciding the content of their teaching. For example, if sustainable development or EE are not seen as essential themes in teaching, they do not necessarily have to be implemented [13,43–48]. Therefore, the activities of education organizations must also be viewed critically in the light of entrepreneurship and sustainable development.

Based on our results and their analysis, we propose the following practices for teacher education and curricula. First, EE and sustainable development would be explicitly addressed in teacher education curricula, its course descriptions, objectives, contents, and pedagogy, thus promoting entrepreneurial, e.g., critical pedagogy [36], and emancipatory pedagogy [10,19]. Second, it would be essential for students to have their own experiences of utilizing this type of education and pedagogy. These experiences affect what they teach, for example, in primary education and how.

While we would expect the Nordic model of education to promote e.g., entrepreneurship and sustainable education, e.g., [49,50], it seems that the importance of these themes has not been understood in-depth in the Nordic countries, and they do not emerge as clearly expressed in these three teacher education curricula in this case study. It seems that the Nordic countries are not significant forerunners in these goals, even if it could have been assumed.

Is it possible that the default is that educational institutions and teachers are sufficiently vigilant and therefore not sufficiently aware of the issue? Or maybe the teaching is done despite the curricula? Although the importance of cross-cutting educational themes is recognized, EE, for example, has also given rise to a wide range of debates. There have been discussions, e.g., of how EE is only related to business activities, even though EE promotes the skills needed in working life on a large scale [66]. Could this have caused some confusion in the Nordic teacher education? However, one should note that the corresponding education debate is not relevant to sustainable education. Therefore, at least not across the board, this conclusion about the background factors influencing the results of this study is not entirely relevant.

We could find out about this in the future studies, e.g., with surveys and interviews to develop our understanding of the phenomenon. The idea on integrating pro-environmental/sustainable and EE could be promoted in the future more explicitly, in which these interdisciplinary educational themes are taken into account more strongly in the curricula. In addition, it would be interesting to share experiences from these practices and at the same time seek models for so-called good practices, and developing communicative networks for teacher educators in the Nordic and global context which may help to push teacher education forward in terms of sustainable and entrepreneurial education. In the future, we could also include other Nordic countries, such as Norway and Denmark. On the other hand, we could deliver a broader international study. Undoubtedly, case and action studies would gather interesting information from these activities; these could be used to further accumulate interesting teaching practices, for example in teacher education. These processes could provide a meaningful basis for further curricula development. Furthermore, an analysis of the relevant global and national policy documents could provide insights into how these competences have transformed, and how these policies may be further improved to support sustainable and entrepreneurial education development.

## 7. Conclusions

In a changing modern era, a developed ability for creativity is an important attribute [67]. Today creative and innovative competencies are often put in relation to the climate crisis and to sustainability. Caiman and Lundegard [68] argue for the need of an education that supports and stimulates creative processes that can serve as a tool in the creation of a more sustainable world. Maybe it is not enough mainly to teach children many facts about sustainability. According to Hedefalk [69], starting from a problem and instead engaging the children in finding solutions can stimulate children and students to create an understanding, for example, of the underlying conflicts and interests that may have caused an environmental impact. Processes of innovative thinking, such as creating new and imagining things that do not exist, stimulate courage and belief that the future can be influenced and changed. Thinking about the potential of entrepreneurship and sustainable education, we wanted to find out how these types of issues have been taken into account in teacher education. We wanted to focus in this way initially on curricula, because by default, the content and activities of teaching are based on them.

In this research, we have studied how Nordic (Finnish, Swedish, and Icelandic) primary teacher education curricula involve entrepreneurial, sustainable, and pro-environmental education. According to our results, EE and sustainable development are taken into account to some extent in curricula. Deliberating how much of these themes are discussed today, we conclude that these educational goals and contents are limited. However, considering the scale of the phenomenon, this is a very small opening to explore the theme only from a few curricula point of view.

We have shown that multidisciplinary research on a theme in one study has its challenges. Nevertheless, on the other hand, adopting this approach has its advantages: EE and sustainable development education have much in common. For example, education for sustainable development (ESD) may be seen as a necessary journey towards sustainable education (SE), and EE could be a part of constructive steps towards sustainable education. EE includes affordances that contribute to inculcating analytical and creative competences. These competences are needed for responsible actions and provide cognitive elements (knowledge and understanding), emotions (identifying needs and problems) and action [19,40]. Taken together, we need entrepreneurial competencies to promote sustainable development. Our research, combining different entry angles, is a significant case opening in studying how to solve global problems by human thinking and behavior.

While our approach, which is built on the broader framework documents, emphasizes the complementarity of sustainable and entrepreneurial development, there may be tensions and even contradictions (e.g., development of enterprises built on wasteful consumption). A more solid justification for how these concerns might be addressed, as

broadly as possible, and how more complementary models might be fostered would be beneficial. Critical thinking, creativity, teamwork, and the study of "real-world problems," as in phenomenon-based learning (29), could also be effective in addressing sustainability and entrepreneurship. In fact, they might be already occurring, as we previously discussed, or, conversely, are they somehow squeezed out or distorted by an emphasis on other priority areas within the respective curricula?

On the other hand, it is good to remember that EE as a concept is also good to transport in the development of the education system consciously. EE is intended to develop society in the direction of both developing and continuing entrepreneurship. If other concepts replace this concept, there is a risk that the aims and dimensions of entrepreneurship education will be blurred.

Our pilot study gives preliminary indicators into where education is moving onto. The results show that education does not respond adequately to societal hashes and even crises such as climate change. Therefore, it is also clear that we need more systematic policy guidance on the integration of interdisciplinary themes in education, such as EE and sustainable development. Moreover, we could consider who should take responsibility for developing these essential issues in education. How should this be handled? How is curricula development guided? Guidelines and actions for the future need to be more concise and explicit goals are needed to support these important areas of knowledge and skills. Furthermore, if universities have autonomy reflecting the needs for education from society, how could universities be motivated to such educational issues, if they are so crucial?

Finally summed up, we could also emphasize the importance of teacher education in the development of societies as a whole. Teacher education has a significant impact on educating citizens far into the future. Thinking about the changing world, we argue that student teachers should be prepared with tools and assignments that stimulate their reflections, creativity, and courage, which are important entrepreneurial attitudes to acquire in order to meet the demands of creating education for the future entrepreneurial citizens, where sustainable development is commonplace in their lives. Thinking, for example, of critical issues such as global warming, we should act now. We are currently training teachers who will soon be entering the workforce. These teachers train the whole nation, and their activities have a long-term impact on the future. If we want to influence the entrepreneurial and sustainable thinking of both children and young people, we should take better account of teacher education. This is also a road to implementing global strategies to enhance life-quality and well-being on a large scale.

Although our research opening is small at this stage, we see its added value, especially in how this opens the door to more advanced curricula development, educational activities, the creation of clearer guiding policy documents, and the research in the field [59] to promote entrepreneurial and sustainable education development in societies.

**Author Contributions:** Conceptualization, J.S.-L., S.R.J., M.H.-L. and M.W.; methodology, J.S.-L., S.R.J. and M.W.; validation, J.S.-L., S.R.J. and M.H.-L.; formal analysis, J.S.-L., S.R.J., M.H.-L. and S.E.-B.; writing—original draft preparation, J.S.-L., S.R.J., M.H.-L., S.E.-B.; writing—review and editing, J.S.-L., S.R.J. and M.H.-L. All authors have read and agreed to the published version of the manuscript.

**Funding:** This research received no external funding.

**Institutional Review Board Statement:** Not applicable.

**Informed Consent Statement:** Not applicable.

**Data Availability Statement:** There is no data in this study.

**Conflicts of Interest:** The authors declare no conflict of interest.

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
