# Peer review of "Responding to Global Challenges through Education: Entrepreneurial, Sustainable, and Pro-Environmental Education in Nordic Teacher Education Curricula"

_sustainability, doi:10.3390/su132212808_

Round 1

Reviewer 1 Report

This manuscript addresses significant questions about the extent to which teacher training programs in selected Nordic nations have incorporated material related to sustainability and entrepreneurship into their curricula. The problem is important to an understanding of challenges associated with ensuring that schooling and teachers are adequately equipped to keep pace with major social and environmental developments, with relevance as well to broader literature related to curricular reform. The analysis is well contextualized with reference to priorities established through the UN Sustainable Development Goals, as well as related frameworks through agencies including the EU and OECD. The paper’s main contribution lies in its focus on the gap between high level goals and the implementation of practices to facilitate the attainment of those goals, a problem that has been explored in extensive bodies of literature but not so much with regard to the phenomena covered in the paper.

The analysis addresses a focused research question, providing a generally convincing discussion of the limited progress that appears to have been made in moving prospects to enhance education related to sustainable development and entrepreneurship in school and teacher training curricula. While the analysis is clearly organized and presented, it could be enhanced by addressing in somewhat more detail a few major issues that emerge with respect to the framework as well as the findings presented, as follows:

(i) Discussion moves from a focus on school curricula (at the primary level) to curriculum at the post-secondary level (of primary teacher education programs). It would be useful to add a bit more detail to enable the reader to understand how these are linked or connected, especially since the intermediate (secondary) level, where there may be greater possibilities to address issues or training options related to sustainability and entrepreneurship than in either of the other levels. The discussion and data demonstrate that many of the main objectives of primary education, and training to teach at that level, emphasize foundational learning and pedagogical activities, so might it not be surprising to see that there is little emphasis on higher order phenomena like sustainable development or entrepreneurship?

Issues of this nature might be addressed by expanding the discussion in the results section, which is very brief (less than a page, as well as table in which the relevant or main findings occupy only a small portion of the points presented). While there may be relatively thin data to work from, the main findings are paid less attention in the paper than the background (re: information about curriculum or references to sustainability and entrepreneurship relative to descriptions of the general teacher education programs and curriculum in each of the three cases) but could benefit from the addition of some more specific information, first, about where, explicitly, these topics are covered in the respective curricula and levels (e.g., with reference to what subjects or types of classes).

This would make it possible to provide a stronger sense of both the specific gaps that do exist as well as prospects for places where the desired kinds of education and training might be introduced or augmented. It is not always clear whether the authors are looking for specific types of curricular topics and subjects or for ‘spaces’ within the curriculum, more generally, where these references may be present or absent and, in particular, how these would fit into curriculum at the primary level. This is important in two respects – first, to guide understanding of where these topics are situated in the framework documents (e.g., indicate the extent to which the learning framework to support “environmental, economic and social transformation” in the OECD Education 2030 statement is aspirational as opposed to an expression of mandated curricular directions), and, second, to draw attention more fully - especially in the conclusion - to strategies to address the observed gaps (see point iv, below).

(ii) Slightly greater elaboration of the methods and analytical framework would be useful. In particular, while it is observed (lines 297-301) that the content analysis was informed by a focus on selected relevant terms, it would be helpful to indicate what was done when reference to these terms was found. In what ways did these appear (e.g., general descriptors, specific activities and learning objectives, other?) and how were these analyzed? Similarly, how were the criteria for assessment (lines 302-304) actually employed in the analysis, and what did this reveal?

(iii) While the analysis, drawing from the broader framework documents, highlights the complementarity between sustainable development and entrepreneurship, there are also possible tensions and contradictions between them (e.g., development of enterprises built on wasteful consumption or resource degradation and depletion). Stronger rationale for how these issues might be addressed, as well as for how it may be possible to foster more complementary models, would be useful. Useful reference is made to competencies such as critical thinking, creativity, collaboration and study of ‘real world problems’ which do suggest openings to address sustainability and entrepreneurship. To what extent might it be possible that these issues are in fact already occurring (though in ways that are beyond the scope of the data presented in this paper) or, conversely, how are they being squeezed out or distorted by emphasis on other priority areas within the respective curricula or by other factors such as resource limitations and fiscal pressures? In this regard, the comment (lines 130-1) that entrepreneurship and sustainable development “appear to be just out of reach of the school curricula in most countries” is intriguing but somewhat confusing without a bit more elaboration that would make clear what this means.

(iv) Finally, while there is a strong normative or prescriptive element in the discussion (i.e., that schools and teacher training programs should be addressing the issues of concern), more could be done to integrate this into the analysis and concluding sections. The discussion points, in particular, to the value of critical pedagogy, which is compatible with these points, but does so only in a loose way. Strengthening of this analytical framework would be beneficial, both to guide the analysis and to point in the final part of the paper to the authors’ vision of what specific kinds of training and curricular activities might be added to achieve the desired aims.

In short, the paper offers potentially valuable contributions that could be enhanced by ensuring that the main elements, including clarification of factors that limit as well as facilitate, the development of more adequate strategies and activities to ensure that teaching and learning about sustainable development and entrepreneurship may be incorporated effectively into curriculum at the appropriate levels.

Author Response

REVIEWER 1

This manuscript addresses significant questions about the extent to which teacher training programs in selected Nordic nations have incorporated material related to sustainability and entrepreneurship into their curricula. The problem is important to an understanding of challenges associated with ensuring that schooling and teachers are adequately equipped to keep pace with major social and environmental developments, with relevance as well to broader literature related to curricular reform. The analysis is well contextualized with reference to priorities established through the UN Sustainable Development Goals, as well as related frameworks through agencies including the EU and OECD. The paper’s main contribution lies in its focus on the gap between high level goals and the implementation of practices to facilitate the attainment of those goals, a problem that has been explored in extensive bodies of literature but not so much with regard to the phenomena covered in the paper.

The analysis addresses a focused research question, providing a generally convincing discussion of the limited progress that appears to have been made in moving prospects to enhance education related to sustainable development and entrepreneurship in school and teacher training curricula. While the analysis is clearly organized and presented, it could be enhanced by addressing in somewhat more detail a few major issues that emerge with respect to the framework as well as the findings presented, as follows:

  • Discussion moves from a focus on school curricula (at the primary level) to curriculum at the post-secondary level (of primary teacher education programs). It would be useful to add a bit more detail to enable the reader to understand how these are linked or connected, especially since the intermediate (secondary) level, where there may be greater possibilities to address issues or training options related to sustainability and entrepreneurship than in either of the other levels.

The discussion and data demonstrate that many of the main objectives of primary education, and training to teach at that level, emphasize foundational learning and pedagogical activities, so might it not be surprising to see that there is little emphasis on higher order phenomena like sustainable development or entrepreneurship?

Thank you for these constructive ideas. We have argued these points of view more explicitly in our discussion part (please see Discussion, para. 1, 2).

Issues of this nature might be addressed by expanding the discussion in the results section, which is very brief (less than a page, as well as table in which the relevant or main findings occupy only a small portion of the points presented). While there may be relatively thin data to work from, the main findings are paid less attention in the paper than the background (re: information about curriculum or references to sustainability and entrepreneurship relative to descriptions of the general teacher education programs and curriculum in each of the three cases) but could benefit from the addition of some more specific information, first, about where, explicitly, these topics are covered in the respective curricula and levels (e.g., with reference to what subjects or types of classes).

Thank you. We have described our results more. (please, see Results, para. 1)

This would make it possible to provide a stronger sense of both the specific gaps that do exist as well as prospects for places where the desired kinds of education and training might be introduced or augmented. It is not always clear whether the authors are looking for specific types of curricular topics and subjects or for ‘spaces’ within the curriculum, more generally, where these references may be present or absent and, in particular, how these would fit into curriculum at the primary level. This is important in two respects – first, to guide understanding of where these topics are situated in the framework documents (e.g., indicate the extent to which the learning framework to support “environmental, economic and social transformation” in the OECD Education 2030 statement is aspirational as opposed to an expression of mandated curricular directions), and, second, to draw attention more fully - especially in the conclusion - to strategies to address the observed gaps (see point iv, below).

Thank you for these good ideas. We hope it is fine we have utilized these suggestions in our text (please, see Research Question and Methodology, para. 3)

(ii) Slightly greater elaboration of the methods and analytical framework would be useful. In particular, while it is observed (lines 297-301) that the content analysis was informed by a focus on selected relevant terms, it would be helpful to indicate what was done when reference to these terms was found. In what ways did these appear (e.g., general descriptors, specific activities and learning objectives, other?) and how were these analyzed? Similarly, how were the criteria for assessment (lines 302-304) actually employed in the analysis, and what did this reveal?

Good points. We have developed these descriptions (please, see Research Question and Methodology, para. 5, 7)  

(iii) While the analysis, drawing from the broader framework documents, highlights the complementarity between sustainable development and entrepreneurship, there are also possible tensions and contradictions between them (e.g., development of enterprises built on wasteful consumption or resource degradation and depletion). Stronger rationale for how these issues might be addressed, as well as for how it may be possible to foster more complementary models, would be useful. Useful reference is made to competencies such as critical thinking, creativity, collaboration and study of ‘real world problems’ which do suggest openings to address sustainability and entrepreneurship. To what extent might it be possible that these issues are in fact already occurring (though in ways that are beyond the scope of the data presented in this paper) or, conversely, how are they being squeezed out or distorted by emphasis on other priority areas within the respective curricula or by other factors such as resource limitations and fiscal pressures? In this regard, the comment (lines 130-1) that entrepreneurship and sustainable development “appear to be just out of reach of the school curricula in most countries” is intriguing but somewhat confusing without a bit more elaboration that would make clear what this means.

Thank you. We also utilized these ideas you provided for us (please see Introduction, para 3; Conclusion, para. 4, 5)

(iv) Finally, while there is a strong normative or prescriptive element in the discussion (i.e., that schools and teacher training programs should be addressing the issues of concern), more could be done to integrate this into the analysis and concluding sections. The discussion points, in particular, to the value of critical pedagogy, which is compatible with these points, but does so only in a loose way. Strengthening of this analytical framework would be beneficial, both to guide the analysis and to point in the final part of the paper to the authors’ vision of what specific kinds of training and curricular activities might be added to achieve the desired aims.

We also appreciate these ideas (please, see Discussion, para. 4)

In short, the paper offers potentially valuable contributions that could be enhanced by ensuring that the main elements, including clarification of factors that limit as well as facilitate, the development of more adequate strategies and activities to ensure that teaching and learning about sustainable development and entrepreneurship may be incorporated effectively into curriculum at the appropriate levels.

Thank you for your profound contribution. We have tried our best to develop this paper according to Your suggestions.

Reviewer 2 Report

Many thanks for submitting your manuscript to the Sustainability journal. It is an interesting topic, which I believe relates to the overall scope of the journal. However, I believe that a number of revisions should be made. Please see comments below:

  • I believe that the relationship between entrepreneurship and sustainability could be discussed more clearly from the outset and within section 2 to 2.2. 
  • Additionally, a summary paragraph or two of the whole of section 2, where generalised themes are highlighted, would be beneficial here.
  • Ensure that page numbers are added for direct quotes.
  • Why is a quantitative method not used? Please discuss further, where appropriate. What are the similar studies in this field, in terms of this methodological approach adopted? What is different here with this research?
  • I believe that two sub-headings or notable paragraphs addressing the recommendations for both practice and policy should be included within the conclusion.

Author Response

REVIEWER 2

Many thanks for submitting your manuscript to the Sustainability journal. It is an interesting topic, which I believe relates to the overall scope of the journal. However, I believe that a number of revisions should be made. Please see comments below:

  • I believe that the relationship between entrepreneurship and sustainability could be discussed more clearly from the outset and within section 2 to 2.2. 

Thank You. We have revised the section 2.1. (please see, para 3,4,5)

  • Additionally, a summary paragraph or two of the whole of section 2, where generalised themes are highlighted, would be beneficial here.

Thank you for this suggestion, (please see, 2.4., para 5)

  • Ensure that page numbers are added for direct quotes.

Revised.  Thank You.

  • Why is a quantitative method not used? Please discuss further, where appropriate. What are the similar studies in this field, in terms of this methodological approach adopted? What is different here with this research?

We have edited our text to clarify our methodological choices (please, see Research Question and Methodology, para. 5,6)

  • I believe that two sub-headings or notable paragraphs addressing the recommendations for both practice and policy should be included within the conclusion.

Thank you. We have integrated recommendations for practice and policy into our discussion and conclusion parts of our text (please see Discussion, para. 4; Conclusion, para. 4, 6).  We also considered creating a separate joint paragraph for these issues, but the text's coherence would have suffered at this point.

Thank you for the useful feedback. We have tried our best to revise our paper.

Reviewer 3 Report

The article approaches topics of great interest in relation to current global challenges from the perspectives of entrepreneurial, sustainable and pro environmental education.

Generally speaking, I can say that the paper is well written, with mostly up-to-date references and it was a pleasure to read it.

However, my only concern is related to the real overall novelty and relevance of the conducted research. This is something that remains rather unclear and a little vague. The authors propose a research conducted on only three universities from three different Nordic countries, and this is done by reading and discussing their online curricula. Thus, from a scientific level perspective, the conducted research and its findings may seem a little flimsy, as the current expectations in this area are much higher. Such a research should be conducted on a much larger scale and the digging done on a greater level of details in order to become truly relevant. What are the consequences or the practical implications of the findings? I suggest the authors to extend their research in order to enhance their findings in the Discussion and Conclusion sections.

Author Response

The article approaches topics of great interest in relation to current global challenges from the perspectives of entrepreneurial, sustainable and pro environmental education.

Generally speaking, I can say that the paper is well written, with mostly up-to-date references and it was a pleasure to read it.

Thank you.

However, my only concern is related to the real overall novelty and relevance of the conducted research. This is something that remains rather unclear and a little vague. The authors propose a research conducted on only three universities from three different Nordic countries, and this is done by reading and discussing their online curricula. Thus, from a scientific level perspective, the conducted research and its findings may seem a little flimsy, as the current expectations in this area are much higher. Such a research should be conducted on a much larger scale and the digging done on a greater level of details in order to become truly relevant.

Thank you for the idea. We have argued the basis of our study and methodology in more detail in Research Question and Methodology (see para. 4,7).

What are the consequences or the practical implications of the findings? I suggest the authors to extend their research in order to enhance their findings in the Discussion and Conclusion sections.

Thank You. Please, see our developed Discussion and Conclusion.

Thank you for your contribution!

Reviewer 4 Report

The manuscript addresses an important issue and is well structured. I have just a few suggestions for the improvement of the manuscript.

Page 2, line 75 - Place the square brackets before the punctuation mark (.)

Page 2, line 87 – Correct the citation in the text (Kollmuss & Agyeman 2002) and use both brackets and parentheses to indicate the reference number and page numbers; for example [17] (p. x). It is not clear why this reference is number 65 in the reference list. As the references must be numbered in order of appearance in the text, this should probably be the reference number 17.

In the Methodology section, the authors should provide details on the characteristics used to assess each curriculum (p.7).

The authors need to reconsider the problem of abbreviations in the text. After defining abbreviations (for instance, ESD, SE, EE), use only the abbreviations. Do not alternate between spelling out the terms and abbreviating them.

In the Discussion section, the authors are expected to provide more suggestions regarding the possibilities of integrating sustainable and entrepreneurship education into the teacher education curriculum.

Author Response

REVIEWER 4

Comments and Suggestions for Authors

The manuscript addresses an important issue and is well structured. I have just a few suggestions for the improvement of the manuscript.

Thank you.

Page 2, line 75 - Place the square brackets before the punctuation mark (.)

Corrected. Thanks.

Page 2, line 87 – Correct the citation in the text (Kollmuss & Agyeman 2002) and use both brackets and parentheses to indicate the reference number and page numbers; for example [17] (p. x). It is not clear why this reference is number 65 in the reference list. As the references must be numbered in order of appearance in the text, this should probably be the reference number 17.

Thank You. We have edited these issues.

In the Methodology section, the authors should provide details on the characteristics used to assess each curriculum (p.7).

Thank you. We have revised our Research Question and Methodology (please, see para 4, 5, 7)

The authors need to reconsider the problem of abbreviations in the text. After defining abbreviations (for instance, ESD, SE, EE), use only the abbreviations. Do not alternate between spelling out the terms and abbreviating them.

We have revised those issues. Thank you.

In the Discussion section, the authors are expected to provide more suggestions regarding the possibilities of integrating sustainable and entrepreneurship education into the teacher education curriculum.

Thank you. We have integrated recommendations for practice and policy into our discussion and conclusion part of our text (please see Discussion, para. 4; Conclusion, para. 6). 

Thank you for Your contribution.

Round 2

Reviewer 2 Report

Many thanks for your revised submission to the journal. I can see that there has been careful consideration of the reviewer comments and additional content has been provided. I believe that it strengthens the paper, and I recommend publication.

Author Response

Thank you again for your profound work which supported us to finalize the paper. 

Reviewer 3 Report

Not all the issues I mentioned in order to improve the scientific level of the research were addressed. The conducted research is too narrow in order to be relevant, thus there is still room for improvement in this regard. However, the paper has been slightly improved overall and can be considered for publishing.

Author Response

Thank you again for your work and ideas which also encourage us for further studies.